# Flexible Graphite/PPG Hybrid Composite-Based Resistive Sensor for Sensing Organic Compounds

**DOI:** 10.3390/s20092651

**Published:** 2020-05-06

**Authors:** Do Hun Kim, Yang Soo Lee, Won Kyu Park, Jin Sun Yoo, Changup Shim, Young Joon Hong, Bong Kyun Kang, Dae Ho Yoon, Woo Seok Yang

**Affiliations:** 1Nano Materials and Components Research Center, Korea Electronics Technology Institute, 25, Saenari-ro, Bundang-gu, Seongnam-si, Gyeonggi-do 13509, Korea; amazingcom@keti.re.kr (D.H.K.); yjs0415@keti.re.kr (J.S.Y.); kangbk84@keti.re.kr (B.K.K.); 2School of Advanced Materials Science and Engineering, Sungkyunkwan University, 2066, Seobu-ro, Jangan-gu, Suwon-si, Gyeonggi-do 16419, Korea; koami@skku.edu; 3Nano Material Division, Cheorwon Plasma Research Institute, Cheorwon, Gangwon-do 24047, Korea; wkpark@cpri.re.kr; 4Division of Advanced Materials Engineering, Kongju National University, 1223-24, Cheonan-daero, Seobuk-gu, Cheonan-si, Chungcheongnam-do 31080, Korea; uhbi@naver.com; 5Department of Nanotechnology and Advanced Materials Engineering, Sejong University, Seoul 05006, Korea; yjhong@sejong.ac.kr

**Keywords:** polypropylene glycol, carbon material, swelling, hydrocarbon sensor

## Abstract

Our objective in this study was to investigate a sensor for volatile organic compounds based on a graphite (G)/polypropylene glycol (PPG) hybrid composite (HC) for sensing hybrid elements. The G/PPG HC sensor films for organic-matter detection were successfully fabricated on polyethylene terephthalate (PET) film with a simple blade-coating method. The sensing paste based on G/PPG (1:2) HC showed good dispersibility and stability. In addition, G/PPG HC sensor films with organic compounds showed different thickness changes as a function of the G/PPG ratio because of the swelling effect of the polymer. The observed differences in resistance of the G/PPG HC films corresponded to those of common organic compounds, suggesting that the disconnection of graphite caused by the swollen PPG matrix caused explosive resistance change. Moreover, we evaluated the sensitivity of typical hydrocarbon materials, such as benzene and toluene, in the sensor film as well as petroleum materials without moisture-induced malfunctions. This study could provoke knowledge about superior sensing with cost-effective and easily scalable materials using polymer/graphite composite-based sensors to improve the sensitivity, selectivity, and stability of chemical sensor applications.

## 1. Introduction

With the recent progress of industries, efforts have been devoted to the resolution of serious medical problems, environmental pollution, and explosion dangers caused by toxic chemical leakage [1,2]. Organic compounds used in a wide array of industrial applications are toxic chemicals with extensive social and environmental significance [3]. They are used either directly or indirectly in many industrial processes. Exposure to them may occur in a range of industrial settings. Because of the deleterious effects of organic compounds used in explosives on human health and the environment, diligent monitoring of organic-compound leakage has become extremely important [4,5]. The development of leak sensors for monitoring toxic chemicals continues to attract widespread interest. There has been an increase in demand for a cost-effective, portable, and highly sensitive device under ambient conditions, which is still a challenging task for the industrial world [6,7,8,9]. Nowadays, leakage sensors are mostly based on shifts in electrical properties that are modulated by interactions with a variety of toxic chemicals [10].

Flexible sensors for organic compounds have a wide range of applications because of their light weight and mechanical flexibility [11,12]. Notably, pipeline defects can cause pipeline leaks, resulting in rapid pressure drops and severe environmental damage. To ensure the smooth functioning of a plant or industry and the safety of the environment, the exact amount and location of pipeline defects must be established [13]. To counter this problem, it would be desirable to use methods such as pipeline detection, vibration vector analysis, pulse-echo methodology, acoustic reflection measurement, transient-wave interceptor interaction and interference detection, probabilistic continuous linear estimators, sound pressure waves, and harmonic ripple analysis. Many studies have reported various methods for detecting oil leaks [14,15,16].

Types of leak sensors that are currently used include polymer, zeolite, carbon nanotube (CNT), infrared, refractive index, and electrochemical leak sensors [17,18]. The polymer can be used for detecting leaks if its material can absorb hydrocarbons and petroleum [19,20]. The sensor is made using CNTs with excellent adsorption of hydrocarbons. A CNT sensor has been developed to improve the sensitivity by using a surfactant [21,22]. Although it is advantageous for polar molecules, it has many errors and is less reliable in a water-based environment. Another way to check for leaks of hydrocarbons is by using an infrared ray, which is also used in qualitative Fourier-transform infrared (FTIR) analysis. It mainly uses an attenuated total reflectance (ATR) method. It has excellent sensitivity and selective reaction [23,24,25]. However, it is difficult to use where environmental variables occur, because of the O-H reaction. Until now, reported sensors for sensing organic compounds have been limited because of their reactivity with water. Sensor malfunction in water-based environments has a serious effect on industrial safety. Therefore, it is requisite to find a method to reduce their reactivity with water [26].

Herein, we present a new type of ultra-sensitive flexible sensor, in which graphite and PPG are used to change the resistance value. PPG is a suitable material that can simply change conductivity because it swells when it encounters an organic solvent. In this paper, we studied the reactivity of PPG with water and organic solvents. The resistance change with various organic compounds was attributed to the deformation and swelling of polymers in the G/PPG hybrid composite (HC) film caused by the disconnection of each G plate. The sensing paste based on G/PPG (1:2) showed good dispersibility and stability.

## 2. Experimental

### 2.1. Materials

Materials of the sensor included graphite, polypropylene glycol (PPG), ethyl cellulose (STD4), and 2-ethoxyethanol. We purchased graphite (QKG-196, 92%) from Samjung C&G (Gyeongsan-si, Gyeongsangbuk-do, Korea), Polypropylene glycol (#2000) and 2-ethoxyethanol (99%) from Samchun Chemical (Seoul, Korea), Ethyl cellulose (STD4) from Dow Corning (Midland, MI, USA).

### 2.2. Sensor Fabrication

A schematic of the sensor-paste formulation process and G/PPG HC film is shown in Scheme 1 and Appendix A. The desired amounts of graphite powder (20 g), STD4 (5.4 g), and 2-ethoxyethanol (100 mL) were added in a 250-mL Polypropylene bottle. Then, we adjusted the G/PPG ratio by changing the amount of PPG. The mixture solution was then pre-homogenized by physical homogenization (SILVERSION L5M) at 7000 rpm for 60 min. Next, the homogenized mixture solution was mixed with a high-pressure homogenizer (Panda Plus 2000, GEA, Düsseldorf, Germany) at 300 bar for 5 min. We fabricated sensor films using a blade coater. These films were dried at 80 °C in air for 10 min. After drying, the sensor was covered by non-woven fabric only when measuring resistance. The non-woven cover can improve reactivity and speed by increasing the G/PPG HC film active area when dropping in small amounts of solvent.

### 2.3. Sensor Characterization

We carried out Raman spectroscopy (Confotec MR520, SOL Instruments Ltd., Minsk, Republic of Belarus), Fourier-transform infrared (FTIR) (M4000, MIDAC Corporation, California, United States), and X-ray diffraction (XRD-6100, SHIMADZU, Kyoto, Japan) analyses. Since the sensor was a paste type, the paste should have been well dispersed. Therefore, we measured the zeta potential (ELSZ-2000, Otsuka, Tokyo, Japan) to measure the dispersion in the paste. We used a contact angle (SEO phoenix-150, Kromtek Sdn Bhd, Kota Kemuning, Malaysia) to measure the response of organic compounds to G/PPG. When evaluating, we measured the film thickness using a Varnier caliber (ABSOLUTE Digmatic Micrometer, Mitutoyo, Sakado, Japan). For the above measuring methods, the sensor was used without the non-woven fabric cover. Finally, we measured the electrical signals (with the non-woven fabric cover) of resistance changes by multi-meter (789 Process Meter, Fluke, WA, United States).

### 2.4. Sensor Measurement

We need to explain how to measure the resistance of the fabricated G/PPG sensor. The Cu wire was connected to the end of the sensor (with the non-woven fabric cover) to connect to a multi-meter (Fluke 789 Process Meter). Both ends of the sensor were connected to a Cu wire and multi-meter. Then, 1 mL of gasoline was dropped onto the sensor as we started to measure the time. Next, we recorded the resistance of the multi-meter every 10 s for 60 s (Appendix A).

### 2.5. Thickness Measurement

We used a sensor without the non-woven fabric cover and measured the thickness using different methods before and after the gasoline reaction. Before the gasoline reaction, the initial thickness was measured by Vernier caliber. After the gasoline reaction, the thickness change was measured by calculating the pixel (px) change rate using contact angle measurement.

## 3. Results and Discussion

Structures of the G/PPG HC with different concentrations of graphite were characterized by ATR–FTIR spectroscopy (Figure 1a). All composites showed different absorption peaks at 2969, 2867, 1452, 1373, and 1220 (asymmetric, symmetric bending vibrations and deformation bands of C–H), and at 1093 (C–O–C stretching vibrations in the –OCH_2_CH_2_– unit) cm^−1^, indicating the formation of PPG [27,28].

The G/PPG HC shows absorption bands between 3000~2800 cm^−1^ and 1500~1250 cm^−1^ originating from C–H stretching and bending vibrations, respectively (Figure 1b) [27]. The strong band at 1155 cm^−1^ can be associated with the C–O–C symmetric stretching vibration in C–O modes caused by ethyl cellulose [29]. In addition, the absorption peak at 1652 cm^−1^ is related to the stretching vibration of the C=C bond in graphite [30]. New absorption peaks were observed at 1155 and 1652 cm^−1^ for the G/PPG HC, suggesting that the graphite powder was successfully incorporated into the PPG.

To investigate whether there was an interfacial interaction between PPG and graphite in the G/PPG HC film, we carried out Raman spectroscopic analysis of the G/PPG HC in the vibration region (800~3000 cm^−1^). Characteristic features of the Raman spectra for pristine carbon can be observed at 1580 cm^−1^ (G band) and 2700 cm^−1^ (2D band) [31]. Raman peaks near 875 and 925 cm^−1^ are caused by extensions to the inside of C-O-C symmetric stretching, and the one near 1455 cm^−1^ is caused by –COO [32,33]. The Raman spectrum of PPG exhibited strong peaks at 2883 and 2939 cm^−1^, corresponding to stretching vibrations of the alkyl chains [34,35]. In addition, peaks associated with the G/PPG HC occurred at 1352, 1567, and 2678 cm^−1^, reflecting D (sp_3_), G (sp_2_), and 2D bands, respectively, caused by graphite [36]. Two sharp peaks at 2880 and 2934 cm^−1^ corresponding to the C–H stretching vibration of the G/PPG HC were also observed. This band slightly shifted about ~3 cm^−1^ from its original position (at 2883 and 2939 cm^−1^ in pure PPG) because of the formation of hydrogen-bridge bonds between the -OH group of graphite and the hydroxyl group of PPG [37].

The zeta potential is the potential between the liquid layer adjacent to the solid phase and the liquid layer constituting the bulk liquid phase. It is a measure of the magnitude of the electrostatic repulsion or attraction between particles. We measured the zeta potentials of the PPG and graphite particles to investigate the affinity between them in the G/PPG HC paste from the point of view of colloidal chemistry (Appendix A). The zeta potential measured for graphite particles in 2-ethoxyethanol was 1166 mV. However, the zeta potential values for graphite particles as a function of the G/PPG ratio (2:1, 1:1, 2:1) steadily increased with the G/PPG ratio. The zeta potential of the G/PPG (1:2) HC paste was +1237.82 mV, indicating that this paste had a more suitable dispersion than other ratios of G/PPG HC paste.

The plausible sensing mechanism of the G/PPG HC film for sensing organic compounds is schematically shown in Figure 2a. A well-aligned G/PPG HC film (25 μm) was fabricated by blade coating, indicating that the graphite was well dispersed in the PPG matrix with sufficient dispersion stability. When the sensor including a G/PPG HC was exposed to organic compounds, a radical chemical reaction occurred. The swelling of the G/PPG HC film under the influence of organic compounds led to a volume phase transition with mechanical stress and separation between the graphite. The volume phase transition and separation of each graphite in the polymer matrix play an important role in detecting the fluctuation of resistance in the G/PPG HC film. In addition, the evaporation of the PPG with organic compounds was observed; it exposed the graphite on the surface of film during the swelling. Figure 2b shows time-dependent thickness changes of the hybrid composite films based on various G/PPG ratios upon exposure to gasoline for 60 s. Because the resistance change response might be associated with volume changes, swelling ratios of the G/PPG sensor film in various formulations were calculated.

The as-fabricated dispersed graphite without PPG did not show any obvious thickness change, and the G/PPG (2:1) and G/PPG (1:1) HCs showed a slight thickness change when they were exposed to gasoline. However, this slight thickness change is not sufficient for the resistance change ratio. In contrast, Figure 2b clearly shows a 15% increase in the thickness change for a G/PPG (1:2) HC according to Equation (1). It is apparent that the observed thickness changes result from the swelling during 10 s. In addition, after 10 s, thickness decreased owing to the decomposition of the polymer matrix upon exposure to the organic compound (Appendix A).
(1)$$\Delta \mathrm{Thickness}\left(\%\right)=\frac{Thickness_{after}-Thickness_{before}}{Thickness_{before}}\times 100$$

Figure 3a–c and Appendix A compare the sensor response of intrinsic responsivity between graphite and G/PPG (2:1, 1:1, 1:2) HC sensors toward 1 mL of water, gasoline, diesel, and kerosene liquids at room temperature during the 90 s. Baseline resistance of the film (30 cm) at room temperature varied from 70 to 150 kilo ohms, depending on the proportion of thickness present and how effectively the electrode gap was bridged.

Data were plotted according to ΔR Equation (2), where R_before_ was the resistance value just prior to the organic solvent exposure, and R_after_ was the resistance value observed during the organic material exposure. As shown in Figure 3a, there was no change in the relative electrical resistance change (ΔR) of the graphite sensors upon exposure to organic materials (diesel and kerosene) for 90 s at room temperature. Only gasoline had a slight ΔR variation. This suggests that the graphite short circuit is not enough to be generated solely by the interfacial reaction of the organic solvent and graphite. As shown in Figure 3b, the ΔR of the G/PPG (1:2) HC film rapidly increased with exposure to organic solvents (gasoline, diesel, and kerosene). It was observed that the increased ΔR of the composite with organic solvents reached a value of about 35,000%~45,000% and was saturated within 10 s. The sensor based on G/PPG (1:2) had outstanding sensitivity compared to that of graphite, G/PPG (1:1) and (2:1). The resistance increase was attributed to the free volume change of the PPG polymer matrix as well as the gap between the electrical junction. The conducting graphite increases and widens because of the ratio of the swelling PPG matrix. The saturation phenomenon arises from the excess range of the fluke 789 process (~45 MΩ). Moreover, the ΔR of the G/PPG (1:2) HC film did not respond to water. These results indicate that G/PPG (1:2) HC film could be applied without the risk of moisture-induced malfunction in an external environment. Figure 3c shows the summarized ΔR variation of the graphite and G/PPG (2:1, 1:1, 1:2) HC sensors. As the PPG content increased, the ΔR variation and reactivity of the G/PPG HC sensors increased. The G/PPG (1:2) HC sensor showed high ΔR variations and saturation to all organic solvents, including gasoline, diesel, and kerosene. However, the reactivity of the G/PPG (1:3) HC increased and the HC film separated easily on the PET film because of physical resistance (Appendix A). In addition, the sensor based on G/PPG (2:1) and (1:1) HCs had slightly lower ΔR variation and reactivity in the diesel.
(2)$$\Delta R\left(\%\right)=\frac{R_{after}-R_{before}}{R_{before}}\times 100$$

We investigated the sensor response by measuring fractional changes in resistance upon exposure to a set of 14 chemically diverse solvents: ethanol, methanol, acetone, 2-ethoxyethanol, cyclohexane, styrene monomer, N-methyl-2-pyrrolidone (NMP), di-methyl-formamide (DMF), isopropyl alcohol (IPA), toluene, ethyl acetate (EA), benzene, and methyl ethyl ketone (MEK). Response curves representing the change in resistance upon exposure to each organic compound are shown at Appendix A. We confirmed that it could be used as a sensor for most hydrocarbons and alcohols.

To account for the apparent relationship of the swelling effect to volume change and sensor activity, we investigated the volume changes of G/PPG (1:2) HC film during gasoline adsorption with a contact-angle camera (Figure 4a–c, Appendix A). We found that the thickness of G/PPG (1:2) HC decreased after the adsorption of gasoline. These results indicated the dilation of the polymer network, confirming the swelling effect of G/PPG (1:2) HC. In addition, the exposed graphite on the top of the G/PPG (1:2) HC film suggested that PPG is dissolved and washed away by gasoline. The recycle test of the G/PPG (1:2) HC film using swelling as a sensing medium is another key reason to consider it for practical chemical sensing applications. However, the ΔR value decreased significantly after only one or two reuses (Figure 4d). The variation in resistance change gradually decreased as the test was repeated. This result indicates that the reaction with gasoline can reduce the swelling effect because of the absence of the removed PPG. In addition, the variation in resistance by water was constant (Figure 4e). Furthermore, the Raman spectra showed that the intensity of C-H bonds at PPG decreased, and air peaks appeared because of the reaction between gasoline and PPG, in good agreement with the exposed G and recycle test shown in Figure 4f.

## 4. Conclusions

In summary, G/PPG HC films were successfully prepared and applied to organic compound sensors using a simple blade-coating method. We confirmed that the graphite in PPG solution was well dispersed and that graphite and PPG formed a hybrid composite structure. As a function of graphite and PPG ratio, the optimized G/PPG (1:2) HC had no chance of malfunctioning in water and had a very high sensitivity to several organic compounds. Based on our characterization outcomes, we attributed resistance change with various organic compounds to the deformation and swelling of the polymer in the G/PPG HC films caused by each disconnected G plate. To the best of our knowledge, cost-effective and easily scalable sensors based on hybrid materials, including polymer/graphite composites, can be conducted with superior chemical sensing applications by improving their sensitivity, selectivity, and stability.

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
