# Peer review of "Flexible Graphite/PPG Hybrid Composite-Based Resistive Sensor for Sensing Organic Compounds"

_sensors, 2020, doi:10.3390/s20092651_

Round 1

Reviewer 1 Report

The authors present a study of easily-prepared graphite/polypropylene glycol (G/PPG) composites, and their performance in the resistive sensing of organic compounds. Significantly larger resistance changes are observed for the G/PPG composites as compared to graphite alone, which the authors attribute to a swelling mechanism.

While I do see the value of simple and inexpensive sensor materials such as the authors target here, overall I think that there are significant issues with this study that make it unsuitable for publication in its current form.  Specific comments are detailed below.

My most serious concerns relate to the lack of sufficient experimental detail. Not only would I be unable to reproduce most of this work based on the level of detail provided, but these information gaps make it impossible for me as a reviewer to fully evaluate the authors' claims. For example:
- From the images provided, it is clear that the G/PPG films on plastic substrate are quite large. In the manuscript text, the authors describe that "resistance of the film (30 cm) at room temperature" was measured, and in the experimental section the model of multimeter used is given. But this is the only experimental detail we have about the resistive sensing measurements that are the core of this study. Are the probes of the test leads held by hand at the ends of the film strips throughout the duration of the exposure? Or is there a more automated and reproducible probe station setup? 
- How are the exposures to organic solvents conducted?  The only information given is that 1 mL is used for 90 sec.  How is this 1 mL applied?  Is the film dipped into the solvent? Is it a vapor phase exposure? Is the solvent dispensed onto the film using a syringe or pipet (as the illustration in figure 2a suggests), and if so, how uniformly is the film covered?  How quickly is the 1mL applied? If the film is uniformly wetted, does this impact the electrical contacts?  If evaporation of volatiles is important to the sensor response and (lack of) recoverability, are these tests conducted in a chamber of well-defined volume, or under an open atmosphere?
- Can more information be provided to better quantify sensitivity?  Yes, the changes in resistance are large, but 1 mL of analyte is a VERY large amount. And, given that it's unclear how the exposure to analyte is actually conducted, it's impossible to compare the sensitivity of these composite materials to other benchmarks in the field. I don't think the authors have any basis to claim "outstanding sensitivity" based on the data provided here.

I'm also unsure about the data that supports the proposed swelling mechanism, because film thickness substantially decreases by the 10-second mark. Nothing about this "confirms the swelling".  Also, all data suggests the PPG is removed upon gasoline exposure (the authors claim via evaporation, but it could also just be washed away by the organic solvents). This seems to leave only the graphite behind on the surface, but then shouldn't the film conductivity be higher once the PPG is removed and a pure graphite surface remains?  There is an inconsistency here that I don't understand.
- related to the thickness measurements, the authors claim to use a digital micrometer for this: is a single point of the film measured by hand every 10 sec, to obtain the data in Fig 2b?  How many different points of the same film were measured? How many measurements at each time point contribute to the error bars shown?  (Since we also don't know exactly how the exposures to gasoline are conducted, we don't know how uniform the changes in thickness should be).
- The Fig S5 contact angle camera video does not work properly in the SI file that I was provided, so I cannot evaluate if this data better supports the swelling hypothesis.

Author Response

The authors present a study of easily-prepared graphite/polypropylene glycol (G/PPG) composites, and their performance in the resistive sensing of organic compounds. Significantly larger resistance changes are observed for the G/PPG composites as compared to graphite alone, which the authors attribute to a swelling mechanism.

While I do see the value of simple and inexpensive sensor materials such as the authors target here, overall I think that there are significant issues with this study that make it unsuitable for publication in its current form.  Specific comments are detailed below.

My most serious concerns relate to the lack of sufficient experimental detail. Not only would I be unable to reproduce most of this work based on the level of detail provided, but these information gaps make it impossible for me as a reviewer to fully evaluate the authors' claims. For example:

- From the images provided, it is clear that the G/PPG films on plastic substrate are quite large. In the manuscript text, the authors describe that "resistance of the film (30 cm) at room temperature" was measured, and in the experimental section the model of multi-meter used is given. But this is the only experimental detail we have about the resistive sensing measurements that are the core of this study. Are the probes of the test leads held by hand at the ends of the film strips throughout the duration of the exposure? Or is there a more automated and reproducible probe station setup?

Response: Thank you for comments. I agree with reviewer’s comments. So, I added the experiments part to test method such as the "2.3. Sensor measurement" and uploading the Video 1. camera video file. This information is added in the revised manuscript.

 <Modifications>

2.3. Sensor measurement

We explain to measure of resistance the fabricated G / PPG sensor. The Cu wire was connected to the end of the sensor (with non-woven fabric) to connect to a multi-meter (Fluke 789 Process Meter). Both end of the sensor was connected to Cu wire and multi-meter. Then 1 mL of gasoline was dropped on the sensor and same time starting to measured time. Next, we record a resistance of the multi-meter every 10 sec during 60 sec (Video 1).

Video 1. Video of measure to resistance the fabricated G / PPG sensor with non-woven fabric.

- How are the exposures to organic solvents conducted?  The only information given is that 1 mL is used for 90 sec.  How is this 1 mL applied?  Is the film dipped into the solvent? Is it a vapor phase exposure? Is the solvent dispensed onto the film using a syringe or pipet (as the illustration in figure 2a suggests), and if so, how uniformly is the film covered?  How quickly is the 1mL applied? If the film is uniformly wetted, does this impact the electrical contacts?  If evaporation of volatiles is important to the sensor response and (lack of) recoverability, are these tests conducted in a chamber of well-defined volume, or under an open atmosphere?

Response: We appreciate the reviewer’s comment. We change these incorrected of test method about "Experiments part" and manuscript. This information is edited in the revised manuscript.

<Modifications>

2.2. Sensor fabrication

A schematic of the sensor paste formulation process and G/PPG HC film are shown in scheme 1 and Figure S1. Desired amounts of graphite powder (20 g), STD4 (5.4 g), 2- ethoxyethanol (100 mL) were added into a 250 mL PP bottle. Then G/PPG ratio was adjusted by changing the amount of PPG. The mixture solution was then pre-homogenized by physical homogenization (SILVERSION L5M) at 7000 rpm for 60 min. Next, the homogenized mixture solution was mixed with at high pressure homogenizer (GEA Panda Plus 2000, USA) at 300 bar for 5 minutes. We fabricated sensor films using a blade coater. These films were dried at 80 ℃ in air for 10 min. After drying, the sensor was covered by non-woven fabric only when measuring resistance.

Scheme 1. Manufacturing process of sensitive flexible organic compounds sensor using G/PPG hybrid composite films.

2.2. Sensor characterization

We performed Raman spectroscopy (SOL equipment confotec MR520), Fourier transform infrared (FTIR), and X-ray diffraction (XRD, SHIMADZU XRD - 6100) analyses. Since the sensor is a paste type, the paste should be well dispersed. Therefore, we measured the zeta potential (Otsuka ELSZ-2000) to determine the dispersion in the paste. Contact angle (SEO phoenix-150 contact angle) was used to measure the response of organic compounds to G/PPG. When evaluating, the film thickness was measured using a Varnier caliber (Mitutoyo ABSOLUTE Digmatic Micrometer). In the case of the above measuring method, the sensor used by without non-woven fabric cover. Finally, we measured electrical signals (with non-woven fabric cover), of resistance changes by multi-meter. (Fluke 789 Process Meter).

2.3. Sensor measurement

We explain to measure of resistance the fabricated G / PPG sensor. The Cu wire was connected to the end of the sensor (with non-woven fabric cover) to connect to a multi-meter (Fluke 789 Process Meter). Both end of the sensor was connected to Cu wire and multi-meter. Then 1 mL of gasoline was dropped on the sensor and same time starting to measured time. Next, we record a resistance of the multi-meter every 10 sec during 60 sec (Video 1).

2.4. Thickness measurement

We used sensor without non-woven fabric cover and measure the thickness using different method about before/after the gasoline reaction. Before the gasoline reaction, the initial thickness was measured by Vernier caliber. After the gasoline reaction, the thickness change was measured by calculating the pixel (px) change rate using contact angle measurement.

- Can more information be provided to better quantify sensitivity?  Yes, the changes in resistance are large, but 1 mL of analyte is a VERY large amount. And, given that it's unclear how the exposure to analyte is actually conducted, it's impossible to compare the sensitivity of these composite materials to other benchmarks in the field. I don't think the authors have any basis to claim "outstanding sensitivity" based on the data provided here.

Response: Thank you for comments. I agree with reviewer’s comments. So, I change the sentence such as “The sensor based on G/PPG (1:2) had outstanding sensitivity compared to Graphite, G/PPG (1:1) and (2:1).”. This information is changed in the revised manuscript.

<Modifications>

The sensor based on G/PPG (1:2) had outstanding sensitivity compared to Graphite, G/PPG (1:1) and (2:1).

I'm also unsure about the data that supports the proposed swelling mechanism, because film thickness substantially decreases by the 10-second mark. Nothing about this "confirms the swelling".  Also, all data suggests the PPG is removed upon gasoline exposure (the authors claim via evaporation, but it could also just be washed away by the organic solvents). This seems to leave only the graphite behind on the surface, but then shouldn't the film conductivity be higher once the PPG is removed and a pure graphite surface remains?  There is an inconsistency here that I don't understand.

Response: We appreciate the careful review. As the author mentioned, Fig 2b. and Fig 4a – b. seems that the phenomenon of contraction and removal rather than expansion is emphasized. We are very sorry for the part we have not described in detail in the experimental part.

First, we will re-measure thickness using by contact angle (without non-woven cover). Because, Fig 2b and Fig 4a - b was show that can’t confirms the swelling and measured thickness by digital micrometer. So, Fig 2b and Fig 4a - b will be change.

Second, this manuscript was not announced that using non-woven fabric cover for measuring resistance. However, we using the non-woven fabric cover. It is suggested that a sensor using a non-woven cover can suppress the PPG removal phenomenon. So, Figure 3 suggests that it is reasonable to increase the resistance within 10 seconds.

So, to deliver the message clearly to readers, the experimental and Fig 2b and Fig 4a - b. were rewritten in the revised manuscript, as below;

<Modifications>

Figure 2b shows time dependent thickness changes of hybrid composite films based on various G/PPG ratios upon exposure to gasoline for 60 sec. Because the resistance change response might be associated with volume changes, swelling ratios of the G/PPG sensor film in various formulation recipe were determined.

The as-fabricated dispersed graphite without PPG failed to show any obvious thickness change and G/PPG (2:1), G/PPG (1:1) shows a slight thickness change when they were exposed to gasoline. However, this slight thickness change is not sufficient for resistance change ratio. In contrast, Figure 2b clearly shows 15 % increase in the thickness change for a composite film of PPG according to Eq.1. It is apparent that observed thickness changes are due to swelling during under 10 sec. In addition, after 10 sec, thickness is decrease owing to decomposition of the polymer matrix upon exposure to the organic compound (Figure S2).

 * 100% (Eq. 1)

Figure 2 Swelling mechanism of G/PPG HC sensor films. (a) Sensor operating mechanism and (b) thickness variation of films at gasoline.

To account for the apparent relationship of swelling effect with volume change and sensor activity, we investigated volume changes of G/PPG (1:2) HC film before and after gasoline adsorption by contact angle camera (Figure 4a-c, Figure S6, Video 2). It was found that the thickness of G/PPG (1:2) HC decreased after the adsorption of gasoline. These results indicated the dilation of polymer network, confirming the swelling effect of G/PPG (1:2) HC. In addition, the exposed graphite on the top of G/PPG (1:2) HC film suggested evaporation and desorption of PPG in gasoline. The recycle test of G/PPG (1:2) HC film using swelling mechanism as a sensing medium is another key criterion to consider it for practical chemical sensing applications. However, ΔR value decreased significantly after one time and two times of reuse (Figure 4d). The variation in resistance change gradually decreased as the test was repeated. This result indicates that the reaction with gasoline can reduce the swelling effect due to the absence of the removed PPG. Furthermore, the Raman spectra showed that the intensity of C-H bonds at PPG decreased and air peak appeared due to reaction between gasoline and PPG, in well agreement with exposed G and recycle test shown in Figure 4e.

 Figure. 4 Swelling phenomenon of (a) bare, (b) after 5sec and (c) after 60 sec reacted G/PPG (1:2) HC film with gasoline. (c) sensitivity of sensor and (d) Raman spectra of G/PPG (1:2) HC film during cycle tests.

- related to the thickness measurements, the authors claim to use a digital micrometer for this: is a single point of the film measured by hand every 10 sec, to obtain the data in Fig 2b?  How many different points of the same film were measured? How many measurements at each time point contribute to the error bars shown?  (Since we also don't know exactly how the exposures to gasoline are conducted, we don't know how uniform the changes in thickness should be).

Response: We appreciate this valuable reviewer’s comment. We measured thickness by digital micrometer for every 10 sec at 5 sample (5 point/sample). However, Fig 2b was not definite data owing to gasoline was not perfectly removed. In that case, G/PPG HC was easily deformed by external force during measuring thickness.

So, we change these incorrected of measuring thickness method from digital micrometer to contact angle measurement. Since the contact angle measurement was recorded in real time after the gasoline reaction, the volume change rate was determined by pixels (px). Also, to deliver the message clearly to readers, the experimental and Fig 2b. were modified and add new sentences in the revised manuscript, as below;

 <Modifications>

2.4. Thickness measurement

We used sensor without non-woven fabric cover and measure the thickness using different method about before/after the gasoline reaction. Before the gasoline reaction, the initial thickness was measured by Vernier caliber (Mitutoyo 227-201). After the gasoline reaction, the thickness change was measured by calculating the pixel change rate using contact angle measurement.

Figure 2b shows time dependent thickness changes of hybrid composite films based on various G/PPG ratios upon exposure to gasoline for 60 sec. Because the resistance change response might be associated with volume changes, swelling ratios of the G/PPG sensor film in various formulation recipe were determined.

The as-fabricated dispersed graphite without PPG failed to show any obvious thickness change and G/PPG (2:1), G/PPG (1:1) shows a slight thickness change when they were exposed to gasoline. However, this slight thickness change is not sufficient for resistance change ratio. In contrast, Figure 2b clearly shows 15 % increase in the thickness change for a composite film of PPG according to Eq.1. It is apparent that observed thickness changes are due to swelling during under 10 sec. In addition, after 10 sec, thickness is decrease owing to decomposition of the polymer matrix upon exposure to the organic compound (Figure S2).

 * 100% (Eq. 1)

Figure 2 Swelling mechanism of G/PPG HC sensor films. (a) Sensor operating mechanism and (b) thickness variation of films at gasoline.

- The Fig S5 contact angle camera video does not work properly in the SI file that I was provided, so I cannot evaluate if this data better supports the swelling hypothesis.

Response: We checked the Fig S5. The Fig S5 was replaced with a better resolution video.

<Modifications>

Video 2. Video of swelling phenomenon taken by contact angle.

Reviewer 2 Report

A sensor for volatile organic compounds based on graphite /polypropylene glycol hybrid composite as sensing hybrid elements was invested. This work is significant and can guide the development of gas sensors. So I think it is worth publishing on “Sensors”. However, there are many ambiguous points need to be clarified before acceptance. The comments are as below:

  1. The first area in need of improvement is related to the editorial aspect. It contains a number of typos and grammatical errors. Careful proof-reading is recommended.
  2. It is precise to provide deviation in Fig.2b, but the data provided in the figure are interlaced and cannot be accurately distinguished. So it need to be improved.
  3. How did the Δ Thickness (%) and the ΔR (%) change after the gas was discharged? More details need to be provided.
  4. In Fig.3b, How did the ΔR (%) change in the first ten seconds?
  5. There are many articles about flexible gas sensors based on resistance measurement, which are worth your learning and reference in the work. For example, Zhang L, et al. Flexible Nanofiber Sensor for Low-concentration Hydrogen Detection. Nanotechnology. 2019; 31: 015504.

Author Response

A sensor for volatile organic compounds based on graphite /polypropylene glycol hybrid composite as sensing hybrid elements was invested. This work is significant and can guide the development of gas sensors. So I think it is worth publishing on “Sensors”. However, there are many ambiguous points need to be clarified before acceptance. The comments are as below:

  1. The first area in need of improvement is related to the editorial aspect. It contains a number of typos and grammatical errors. Careful proof-reading is recommended.

Response: Thank you for your concerning comment and advice about our paper. We attached a certificate of proof that we received English proofreading through an English correction company.

  1. It is precise to provide deviation in Fig.2b, but the data provided in the figure are interlaced and cannot be accurately distinguished. So it need to be improved.

Response: We appreciate this valuable reviewer’s comment. Fig 2b. was not definite data owing to gasoline was not perfectly removed. In that case, G/PPG HC was easily deformed by external force during measuring thickness. So, the data was presented through re-testing by contact angle measurement. Since the contact angle measurement was recorded in real time after the gasoline reaction, the volume change rate was determined by pixels. Also, to deliver the message clearly to readers, the experimental and Fig 2b. were rewritten in the revised manuscript, as below.

<Modifications>

Figure 2b shows time dependent thickness changes of hybrid composite films based on various G/PPG ratios upon exposure to gasoline for 60 sec. Because the resistance change response might be associated with volume changes, swelling ratios of the G/PPG sensor film in various formulation recipe were determined.

The as-fabricated dispersed graphite without PPG failed to show any obvious thickness change and G/PPG (2:1), G/PPG (1:1) shows a slight thickness change when they were exposed to gasoline. However, this slight thickness change is not sufficient for resistance change ratio. In contrast, Figure 2b clearly shows 15 % increase in the thickness change for a composite film of PPG according to Eq.1. It is apparent that observed thickness changes are due to swelling during under 10 sec. In addition, after 10 sec, thickness is decrease owing to decomposition of the polymer matrix upon exposure to the organic compound (Figure S2).

 * 100% (Eq. 1)

Figure 2 Swelling mechanism of G/PPG HC sensor films. (a) Sensor operating mechanism and (b) thickness variation of films at gasoline.

Figure S2. G/PPG HC sensor films thickness variation with error bar at dropped gasoline.

  1. How did the Δ Thickness (%) and the ΔR (%) change after the gas was discharged? More details need to be provided.

Response: We change these incorrected of test method about "Experiments part" and manuscript. This information is edited in the revised manuscript.

<Modifications>

2.3. Sensor characterization

We carried out Raman spectroscopy (SOL equipment confotec MR520), Fourier transform infrared (FTIR), and X-ray diffraction (XRD, SHIMADZU XRD-6100) analyses. Since the sensor is a paste type, the paste should be well dispersed. Therefore, we measured the zeta potential (Otsuka ELSZ-2000) to measure the dispersion in the paste. We used a contact angle (SEO phoenix-150 contact angle) to measure the response of organic compounds to G/PPG. When evaluating, we measured the film thickness using a Varnier caliber (Mitutoyo ABSOLUTE Digmatic Micrometer). For the above measuring methods, the sensor was used without the non-woven fabric cover. Finally, we measured electrical signals (with non-woven fabric cover), of resistance changes by multi-meter (Fluke 789 Process Meter).

2.4. Sensor measurement

We need to explain how to measure the resistance of the fabricated G / PPG sensor. The Cu wire was connected to the end of the sensor (with non-woven fabric cover) to connect to a multi-meter (Fluke 789 Process Meter). Both ends of the sensor were connected to a Cu wire and multi-meter. Then 1 mL of gasoline was dropped onto the sensor as we started to measure the time. Next, we recorded the resistance of the multi-meter every 10 sec for 60 sec (Video 1).

2.5. Thickness measurement

We used a sensor without the non-woven fabric cover and measured the thickness using different methods before and after the gasoline reaction. Before the gasoline reaction, the initial thickness was measured by Vernier caliber. After the gasoline reaction, the thickness change was measured by calculating the pixel (px) change rate using contact angle measurement.

Video 1. Video of measure of the resistance of the fabricated G/PPG sensor with non-woven fabric.

  1. In Fig.3b, How did the ΔR (%) change in the first ten seconds?

Response: We appreciate the careful review. We know the importance of resistance change ratio (ΔR) before 10 seconds. However, resistance increased rapidly, making it difficult to divide in 1 second increments. Therefore, a video is attached to Video 1.

 <Modifications>

Video 1. Video of measure of the resistance of the fabricated G/PPG sensor with non-woven fabric.

  1. There are many articles about flexible gas sensors based on resistance measurement, which are worth your learning and reference in the work. For example, Zhang L, et al. Flexible Nanofiber Sensor for Low-concentration Hydrogen Detection. Nanotechnology. 2019; 31: 015504."

Response: We appreciate the reviewer’s comment. We cited in the manuscript for ["Nanotechnology, 2019; 31 : 015504:]

<Modifications>

The development of leak sensors for monitoring toxic chemicals continues to attract widespread interest. There has been an increased demand for a cost-effective, portable, and highly sensitive device under ambient conditions, which is still a challenging task to the industrial world [6-9].

Zhang L.; Jiang H.; Zhang J.; Huang Y.; Tian J.; Deng X.; Zhao X.; Zhang W. Flexible nanofiber sensor for low-concentration hydrogen detection. Nanotechnology. 2019, 31, 015504

Round 2

Reviewer 1 Report

The revised version of this manuscript is substantially improved compared to the original version, and the authors have overall done a good job of addressing my comments.

The main strengths of the authors' study are:

  • the simplicity and low-cost fabrication of the sensor device
  • the rapid and strong response of the sensor, and its stability/lack of response towards water

The main drawbacks are:

  • lack of reusability (Fig 4d)
  • lack of selectivity (Fig S5)

Overall I think that the manuscript should be published after minor revisions. There is some language in the newly-added text that should be edited to be more professional (for example, section 2.4 "We need to explain how to measure..."). The purpose of the fabric cover should be better explained in the manuscript text.  And I think that "evaporation" is probably not an accurate word to describe what happens to the PPG (it seems more likely that the PPG is dissolved and washed away by the organic liquid). 

Given how strong the observed responses are, I think an interesting follow-up study would be to look at the response to gasoline vapor rather than liquid -- it seems reasonable that a response could be observed at fairly low concentrations, which would increase the potential utility of these sensors in real-world settings for early detection of trace leaks.

Author Response

The main strengths of the authors' study are:

the simplicity and low-cost fabrication of the sensor device the rapid and strong response of the sensor, and its stability/lack of response towards water

The main drawbacks are:

lack of reusability (Fig 4d)

lack of selectivity (Fig S5)

Response: Thank you for your concerning comment and advice about our paper. We change these lack part about reusability and selectivity. This information is added in the revised manuscript.

<Modifications>

However, ΔR value decreased significantly after only one or two reuses (Figure 4d). The variation in resistance change gradually decreased as the test was repeated. This result indicates that the reaction with gasoline can reduce the swelling effect because of the absence of the removed PPG. In addition, the variation in resistance by water was constant (Figure 4e). Furthermore, the Raman spectra showed that the intensity of C-H bonds at PPG decreased and air peaks appeared because of the reaction between gasoline and PPG, in good agreement with the exposed G and recycle test shown in Figure 4f.

 Figure 4. Swelling phenomenon of (a) bare, (b) after 5 sec and (c) after 60 sec reacted G/PPG (1:2) HC film with gasoline, (d) Sensitivity of sensor, (e) sensor stability in water and (f) Raman spectra of G/PPG (1:2) HC film during cycle tests.

Figure S5. (a) - (d) Sensitivity of G/PPG (1:2) HC film with various liquid organic compounds.

Overall I think that the manuscript should be published after minor revisions. There is some language in the newly-added text that should be edited to be more professional (for example, section 2.4 "We need to explain how to measure..."). The purpose of the fabric cover should be better explained in the manuscript text.  And I think that "evaporation" is probably not an accurate word to describe what happens to the PPG (it seems more likely that the PPG is dissolved and washed away by the organic liquid).

Response: We appreciate the careful review. So, I added the purpose of using fabric cover. Also, corrected the incorrect "evaporation" proposal. This information is edited in the revised manuscript.

<Modifications>

2.2. Sensor fabrication

A schematic of the sensor-paste formulation process and G/PPG HC film are shown in Scheme 1 and Figure S1. The desired amounts of graphite powder (20 g), STD4 (5.4 g), and 2-ethoxyethanol (100 mL) were added in a 250-mL PP bottle. Then we adjusted the G/PPG ratio by changing the amount of PPG. The mixture solution was then pre-homogenized by physical homogenization (SILVERSION L5M) at 7000 rpm for 60 min. Next, the homogenized mixture solution was mixed with a high-pressure homogenizer (GEA Panda Plus 2000, USA) at 300 bar for 5 minutes. We fabricated sensor films using a blade coater. These films were dried at 80℃ in air for 10 min. After drying, the sensor was covered by non-woven fabric only when measuring resistance. The non-woven cover can be improve reactivity and speed by increasing of G/PPG HC film active area when dropping in small amounts solvent.

These results indicated the dilation of the polymer network, confirming the swelling effect of G/PPG (1:2) HC. In addition, the exposed graphite on the top of G/PPG (1:2) HC film suggested that PPG is dissolved and washed away by gasoline.

Given how strong the observed responses are, I think an interesting follow-up study would be to look at the response to gasoline vapor rather than liquid -- it seems reasonable that a response could be observed at fairly low concentrations, which would increase the potential utility of these sensors in real-world settings for early detection of trace leaks

Reviewer 2 Report

The work has been improved significantly and the test methods have been described adequately with more details suppled. So,  I think the manuscript warrants publication in Sensors now.

Author Response

Thank you for your concerning comment and advice about our paper. We appreciate the care of your review.